# Water–Energy Nexus in Typical Industrial Water Circuits

**Miguel C. Oliveira [1], Muriel Iten [1,\*], Henrique A. Matos [2]**  **and Jochen Michels [3]**

[1]  Low Carbon and Resource Efficiency, R&Di, Instituto de Soldadura e Qualidade, 4415 491 Grijó, Portugal; dmoliveira@isq.pt

[2]  Departamento de Engenharia Química, Instituto Superior Técnico, 1049-001 Lisboa, Portugal; henrimatos@tecnico.ulisboa.pt

[3]  Department, Dechema Research Institute, 60486 Frankfurt am Main, Germany; Jochen.michels@dechema.de

\*  Correspondence: mciten@isq.pt; Tel.: +351-22-747-1950

**Abstract:** Water–energy nexus has been recognized as an important and challenging issue, namely in industry. This is due to industry reforms, increasing demand, and climate change. This concept focuses on the link between energy and water infrastructure. Overall, there is limited understanding of the nature of this link, as it is assumed that water is not a threat to the energy sector or an influence of the electricity to the water resources. This work aims to present and evaluate case studies related to typical industrial water circuits. These circuits represent some of the most relevant industrial sectors in terms of water–energy nexus such as: steel industry, chemical industry, paper and pulp industry, and food industry. Moreover, these sectors also cover typical industrial water circuits, namely: cooling circuit, gas washing circuit, water treatment circuit, transportation circuit, and quenching circuit. The circuits have firstly been assembled in OpenModelica software considering the equipment and physical layout of each circuit. According to their actual operation conditions, the energy and water consumption have been estimated. Furthermore, water and energy efficiency improvement measures have been proposed and implemented into the assembled models. This enabled a techno-economic assessment based on the implementation of the improvement measures. In order to contextualise these results into the industrial trends, the achieved water and energy savings are projected into potential national and sectorial savings considering the current levels of water and energy demand for each sector.

**Keywords:** water–energy nexus; water efficiency; energy efficiency; industrial water circuits

## 1. Introduction

The industrial sector represents a considerable share of both energy and water use in the world. In the European Union, industry corresponds for 25.3% of the final end-use of energy [1]. The European industry accounts for about 40% of the total water abstractions [2], hence such substantial energy and water consumptions lead to the need of increasing energy and water efficiency measures in industry. Several directives have been implemented in order to implement energy efficiency measures at global level, European Union (EU) level or at systems level. One of the first benchmark policy for energy efficiency improvement in EU is the 2005 Green Paper on Energy Efficiency [3], which claimed strengthening energy efficiency policies. It also highlights the interdependence of energy and economic savings and the need to act on energy production and distribution. Nonetheless, it was the European Union Action Plan for Energy Efficiency [4] that encouraged the adoption of innovative technologies in industry and buildings. In its turn, the climate change and energy package of the Europe 2020 Strategy stablished the key parameters to be achieved by the year 2020 namely: the reduction of 20% of the

greenhouse gas emissions relatively to 1990 levels, the share of renewable energy use of 20%, and a 20% improvement of energy efficiency [5]. The 2012 Energy Efficiency Directive [6] established energy efficiency target of the Europe 2020 Strategy, delineating measures to be obliged to a more efficient use of energy from the production to the final consumption. It involved the drawing of National Energy Efficiency Actions Plans (NEEAP's) that establish the estimated energy consumption values and planned improvement measures. Currently, 2030's climate and energy framework [7] establishes tighter targets, namely the achievement of at least 27% improvement of energy efficiency by the year 2030. Looking closely to industrial electricity consumption, the pumping systems are currently related to 20% of the total electricity use [8]. A directive emerged in this regard, establishing the requirements for ecodesign of water pumps aiming to improve the operation of pumps to achieve greater energy savings and persuade the use to more energy-efficient pumps [9]. The high energy consumption of pumps is directly related with water consumption, and, therefore, the overall use of water in the European Union is also considered in directive, such as the Water Framework Directive [10]. Among other concerns, this directive highlighted that the water scarcity in Europe is due to water demand exceeding the available water resources. From the abovementioned information, a new concept has risen named water–energy nexus. It deals with the interdependencies between water and energy consumption [11]. While a summit regarding all the concerns related to water and energy use in the world have been existing for long, the concept was primarily introduced by Gleick [12]. With a systematic link to climate, it defines the concept of the nexus, then the new water paradigm and peak ecological water, as well as the two connections that serve as the base of this concept: water for energy and energy for water. Water–energy nexus is grounded on the evidence that water wastage has direct implications for energy wastage, owing to the sourcing, treatment and distribution of water necessarily requiring the use of great amounts of energy. Figure 1 presents a diagram of the interdependencies of water and energy nexus, encompassing its generation and use from upstream to the downstream.

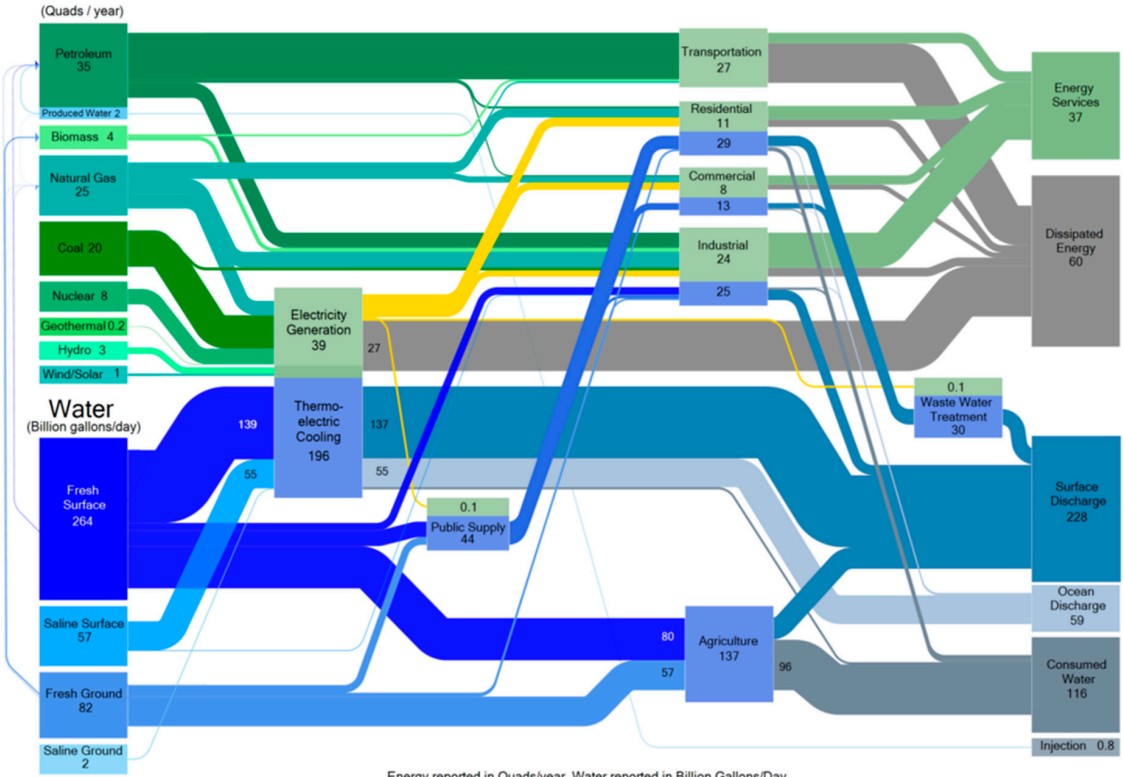

**Figure 1.** Sankey diagram for the water–energy nexus (adapted from [13]).

The contact points between water and energy at an industrial level can be observed in Figure 1. It shows that the electricity used by industries is generated by primary sources, such as natural gas and

coal, as well as by the conversion of nuclear, geothermal, hydro, wind and solar energy. In its turn, the water used in industrial plants are provided by fresh surface water resources. From the downstream of the diagram is it relevant to observe the dissipated energy and the consumed water. The dissipated energy corresponds to the amount of energy that is in excess in a plant and savings may be achieved through the application of improvement measures.

In this alignment, the importance of water–energy nexus of typical industrial water circuits has been identified by the WaterWatt project [14], and in particular through this work. Industrial water circuits consist of a set of process components and unit operations connected by water streams, therefore representing a water circulation system within a plant. Such systems are installed for different needs—cooling, treatment, washing, and transportation—required for the production processes. The WaterWatt project aims to overcome current technical and social barriers to improve the efficiency of water and energy in typical industrial water circuits. For this, a platform has been developed integrating several resources, enabling relevant stakeholders to enhance the energy and water efficiency of water circuits. In this work, the energy-water potential savings have been analyzed for typical industrial water circuits (IWC). For such analysis, different case-studies (part of the WaterWatt project [14]) are considered. These are included in four European countries, namely: Germany, Portugal, United Kingdom and Norway. These case-studies correspond to representative water–energy nexus industrial sectors such as: iron and steel, pharmaceutical, paper and pulp, and food industries. The study also considers the concepts and implications of water–energy nexus, namely the influence of the optimization of industrial water systems on the energy and water consumption. Firstly, eleven case-studies of industrial water circuits are presented. These circuits have been assembled and modelled in OpenModelica through a developed and tailored WaterWatt library. Considering the current status of the IWC, improvement measures have been identified and implemented into the models respectively. This has been followed by a techno-economic assessment considering the identified measures and at the last, the achieved improvements are reflected into energy and water consumption in EU referred countries by projecting potential savings of the water and energy savings at each sector and country level.

## 2. Water–Energy Nexus in Industries at EU Level

The demand for water and energy have been rapidly increasing in the whole world. The supply and demand of water and energy are affected by fluctuations and quality variability, its high consumptions have effects on climate change and the environment. The water–energy nexus deals with the relationship between the use of these utilities and also claims that it is necessary to proceed with water management in the same manner as energy management.

The concerns regarding water used have been regarded almost as important as the ones regarding energy. This prospect lead to need to delineate water management strategies in each part of the world. Therefore, by hypothesis, it has been pointed that similar measures must be applied to the case of water [11]. By neglecting the concern of water supply, the energy supply area is affected, the inverse occurring as well. UN launched the World Water Development Report in 2014 [15], outlining the current situation about water utilization and highlighting the need to implemented approach considering the concerns of both water and energy. In the European Union, it was launched the Water Framework Directive in 2000. In addition to approach the problems raised by human activities regarding water quality, which highlighted the concern of water scarcity in EU [11]. This concern was proposed to be solved by the reduction of water wastage. In the case of the manufacturing industry, the increased energy consumption in plants are due to the increased overall demands by the factories, this is, by both the production processes and the requires systems installed to allow the operation of a plant. The increased demands by the plants require increase in the plant capacity on the supply of important utilities, such as water, which is associated to several industrial uses, such as cooling, washing and transportation. Therefore, it is clear that higher levels of energy use industrially are also associated to a higher water use.

In an industrial plant, water and energy are treated as flows. As stated by Thiele et al. [16], such flows are dynamically interconnected, and as such, it is difficult in practice to analyze the application of improvement measures to different energy and resource flows. Furthermore, Thiele et al. [16] identified the existence of a research gap on water–energy nexus concept application. The study claims that despite the wide availability of diverse approaches on a macro-economic level, approaches at a plant level are still scarce. Nevertheless, the water–energy nexus is also linked to sustainable development. A sustainable design and operation of a plant requires water and energy management approaches. Fouladi and Linke [17] developed an optimization framework for the water and energy efficiency for processes with suppress of energy at various qualities and accounting for different sustainability metrics. Even with a limited contextualization of the water–energy nexus within the manufacturing industry, improvement measures to reduce energy and water consumption at EU context have been explored. Actions to optimize water systems regarding the reduction of electricity consumption have been analyzed by Cabrera et al. [18]. This study proposes a total of eight improvement strategies for the energy efficiency of pressurized water systems. It also refers the typical and expected energy savings achieved with such measures, defining a road map for the such applications. Cabrera et al. [18] claims that in EU the improvement of the pumping water systems can achieve 20–30% of energy consumption reduction. Nowak et al. [19] furtherly contributed to the optimization of pumping systems, describing an optimization method for variable speed pumps and the achieved energy savings. In addition, Annus et al. [20] studied the electricity savings with the application of Variable Speed Drives (VSD's) and the variation of a water system flow rate related to the circulation pumps in domestic household systems.

## 2.1. Overview of the Energy Efficiency Strategies and Frameworks at European Level

The improvement of energy efficiency at European Union is secured by a proactive position by the representatives of the manufacturing industry in each Member State. One of the most relevant actions encompassed by the adoption of this proactive position is the implementation of the Best Available Technologies (BAT) in industry [21]. The responsible bodies for energy management of each Member State publish documents about the effective adequacy of process equipment and operations to new technologies and strategies. The most prominent actions and measures taken in each Member State and sector can be observed in Table 1, as well a general view on energy efficiency. Moreover, Table 2 summarize the most prominent actions and measures considered in each industrial sector, related to the major programs and projects.

**Table 1.** Summary of energy efficiency strategic and technological framework at European Union (EU) member states.

| Member State | Improvement of Energy Efficiency | Measures at an Industrial Level |
|---|---|---|
| Germany [22] | Superior progress compared to the other Member States | Reinforcement of the support by public authorities Development of efficiency financing networks |
| Portugal [23] | Overall positive effect Additional support and training campaigns were appointed | Deployment of the Intensive Energy Consumption Management System |
| United Kingdom [24] | Unappreciable (the efficient use of energy is not treated as an opportunity) | Electricity Demand Reduction pilot (launched with the objective to promote the reduction of energy demand at peak times) |
| Norway [25] | Appreciable (energy supply turned more flexible and the dependence on direct electricity for heating decreased) | Introduction of a pre-project for energy measures, new energy and climate technology Introduction of new technology Formation of a database for energy consumption and production |

**Table 2.** Energy Efficiency Strategic and Technological Framework of each sector.

| Sector | Programs and Projects |
|---|---|
| Chemical Industry | SPIRE (Sustainable Process Industry through Resource and Energy Efficiency)—Public-private partnership program aiming to develop technologies and the best practices sharing to ensure enhanced resource and energy efficiency [26] E4Water project (encompassed by SPIRE [27])—Aims to boost the recycle and reuse of water through the development and testing of innovative materials, process technologies and tools [28] SPICE$^3$ platform (Sectoral Platform in Chemicals for Energy Efficiency Excellence) launched with the objective to support companies in the achievement of energy savings [29] |
| Food Industry | Food Drink Europe's Environment Vision document [30]—Describes the best practices to increase energy efficiency and initiatives related to mandatory energy audits Climate Action and the Food and Drink Industry [31] |
| Paper and Pulp Industry | Two Team Project—Aims to identify breakthrough technology concepts to be adapted and developed for the sector, one of these being the total use of electricity through renewable energies [32] |
| Metal Industry | ULCOS (Ultra Low Carbon Steel)—Group of projects aiming to investigate potential breakthrough technologies to surpass the limitations regarding energy efficiency and gas emissions reductions in the sector, remarkably the development of a blast furnace with top gas recycling [33] The sector also embraces the SPIRE program [34] |

## 2.2. Water and Energy Consumption

In order to investigate the interdependencies between the use of water and energy in the scope of this work, a first insight of the water and energy consumption in each industrial sector and country encompassed by each case study must be considered. The electricity consumption for each industrial sector in Germany and Portugal, as well as the average consumption for the European Union, is presented in Figure 2. The water consumption for the respective sectors and countries is presented in Figure 3. This average consumption was determined per enterprise for each sector and region. Table 3 summarized the used sources for the data on water and energy consumption.

**Table 3.** Sources for energy and water consumption.

| Member State | Data for Energy Consumption | Data for Water Consumption | Data for Number of Enterprises |
|---|---|---|---|
| Germany | | Destatis [36] | |
| Portugal | Eurostat [35] | Associação Empresarial de Portugal [37] | Eurostat [38] |
| European Union | | EUROPA [1] | Yeen and Kantamaneni [39] |

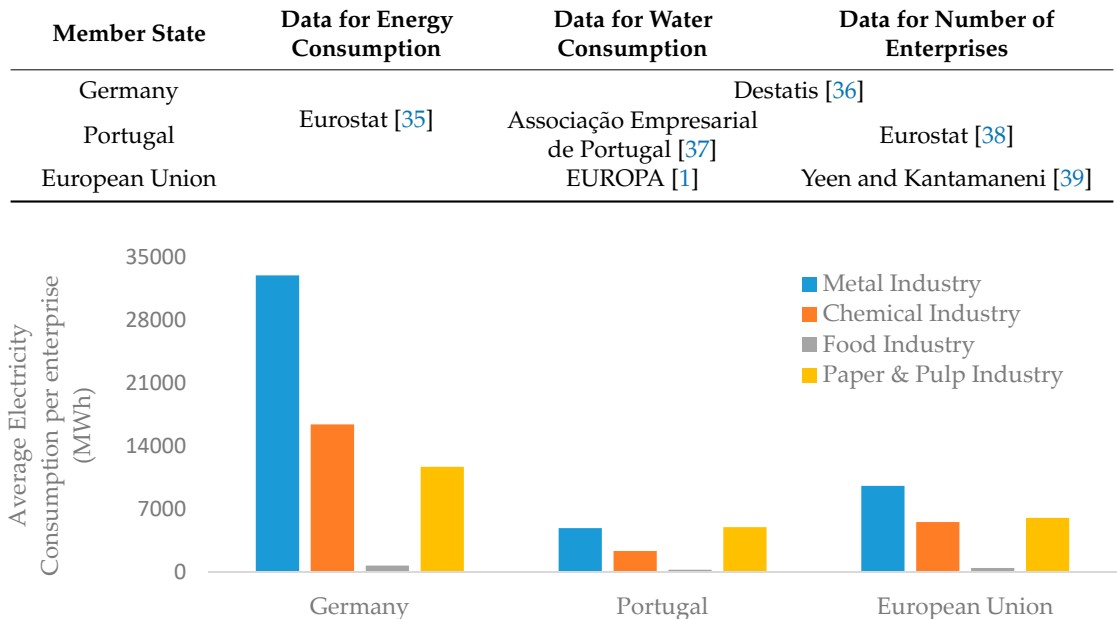

**Figure 2.** Average electricity consumption for each industrial sector and region (adapted from [34,36–38]).

The total water consumption for each industrial sector in the European Union was determined by admitting that the number of consumers was equal to the number of inhabitants in the region.

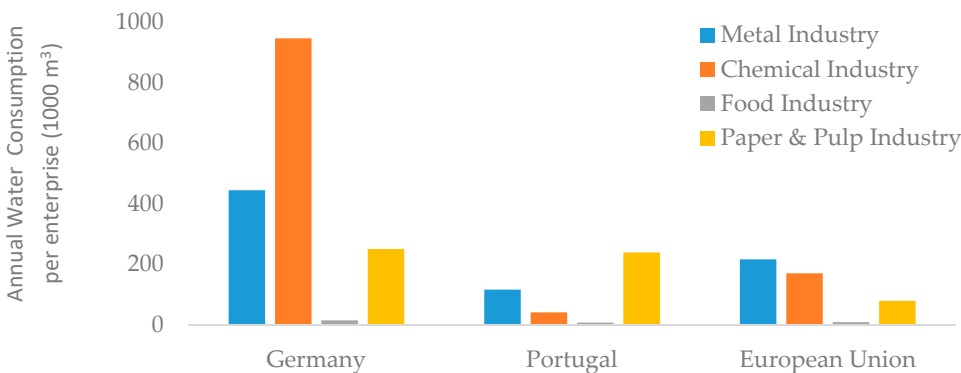

**Figure 3.** Average water consumption for each industrial sector and region (adapted from [1,36,39]).

Figures 1 and 2, shows that the average levels of electricity consumption are nearly aligned with the levels of water consumption. Such evidence is consistent with the observed interdependence between water and energy consumption in industry, as mentioned above. It is to note that the levels of both energy and water consumption in Germany are higher than the consumption levels of Portugal and the average levels of the European Union. The metal industry is generally the higher consumer, followed by the chemical industry and paper and pulp industry. However, it is observed that the level of water consumption for the German chemical industry considerably exceeds the level of the metal industry, which is also observable between the Portuguese paper and pulp industry and metal industry. The food industry is, generally, associated to an average lower energy and water consumption relatively to the other industries, which is essentially due to the high number of enterprises comparatively to the other sectors.

### 2.3. Analysis of Water Pumping Systems

The electric motors account for a considerable share of the total energy consumption in industry. As mentioned above, it is claimed that pumping systems account for about 20% of the total electricity demand in the world [8]. In the European Union, motor-driven systems are reported to correspond for 65% of the total electricity consumption [40].

The pumping systems are an integral part of almost all water systems. Industrial circuits are used for cooling, gas washing, transportation, and in treatment units and all include pumps. The pumping systems present a significant impact on the operation of an industrial circuit, either to fulfil the water and head demands of a circuit or the high share of electricity consumption. The pumping systems account for a significant part of the total electricity consumption in a plant, therefore its performance is crucial and must be secured in order to maintain the levels of energy consumption. Hydraulic efficiency of the pumps over lifetime is compromised due to long operational time, erosion, corrosion, wear, and cavitation occurrence. The refurbishment of pumps corresponds to a regain of 10% of its efficiency during the lifetime of the pumps [41]. On the other hand, the general use of electric motors in the European industry was studied by Saidur [42]. In addition to provide a basis to the analysis of electricity consumption in the European Industry, the author also refers the issues related to the operation of electric motors, such as the types of energy losses. The potential of the use of technological measures for the achievement of energy savings has also been described, namely the use of VSD's and high efficiency motors. Furthermore, Almeida et al. [43] developed a more detailed benchmark to specifically the share of use of electric motors in the industrial sector. The disaggregation of motor electricity consumption by end-use in the industrial sector referred by Almeida et al. [43] is represented in Figure 4.

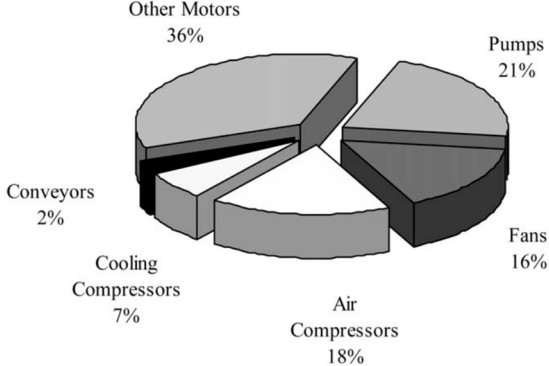

**Figure 4.** Disaggregation of electric motor use in the European Union (adapted from [40]).

Almeida et al. [43] extended the analysis by an extrapolating the data of the use of electric motors from one single country to the EU level. This was applied to the share of use of electric motors relative to the total electricity consumption but also to the share of use of specific electric motors, such as pumps and fans. Table 4 summarizes the estimated values for the shares of energy consumption by electric motors and each type of motor.

**Table 4.** Shares of electricity use by industrial sector in the European Union.

| Sector | Share of Motors Electricity Consumption (%) |
|---|---|
| Paper, pulp and print | 75.1 |
| Food, beverage and tobacco | 89.8 |
| Chemical | 71.9 |
| Iron and steel | 66.3 |
| Other industry | 59.9 |

As it may be observed from Table 4, most of the industrial sectors approached in this work are associated to a consumption share higher than the level of Europe, which is 65% of share. This prospect led to the necessity to elaborate and, furthery, implement improvement measures to optimize the operation of electric motor-driven systems in industry.

## 3. Materials and Methods

The techno-economic assessment for the implementation of improvement measures is performed according to Equation (1), in which *PB* designates the payback period, $C_{Inv}$ the investment costs and *Sav* the economic savings.

$$PB = \frac{C_{Inv}}{Sav} \tag{1}$$

In the scope of this work, a simple payback is determined and therefore no discount rate is considered. Moreover, the electricity cost for each case study is based on each country electricity rates.

The implementation of improvement measures has also the potential to generate savings in water consumption. The reduction of water consumption may be obtained considering the total volume of water in the circuit in the initial and improved scenarios. The volume of water is obtained by the product of the water flow rate and a residence time, according to Equation (2).

$$V_w = Q_w \times t_{res} \tag{2}$$

As the residence time is a measure of the time which a fluid remains in a vessel, for instance, in a water tank, the tanks of a same circuit may, in practice, be associated to different residence times, considering a same water flow rate and different tank capacities. The Equation (2) is, thus, a generic formula for the calculation of water volume in different points of a circuit. The total water volume in a

circuit may, theoretically, be calculated by the sum of all the parcels of the product of water flowrate and residence time for a point of the circuit. In the scope of this work, such procedure was not adopted for the determination of water savings, as the tank models of the library used to create the circuit models does not consider the residence time as a parameter. In order to calculate the water savings in the approached circuits, it was assumed that the residence time is maintained after the adjustment of the circuit operation to lower water demands, this is, the residence times at specific points of the circuits are constant for all the considered operational conditions. Considering this assumption, the decrease of water volume is directly proportional to the decrease of the water volumetric flowrate. In this scope, the savings in water consumption may be determined as the reductions in water flowrate, thus consisting in reductions of water volume in hourly basis.

A set of indicators for the performance of water supply systems was proposed by Vilanova and Balestieri [44]. Although it refers to different systems and circuits, these indicators may be applied for the current case studies, as it is also related with specific components such as water pumps. The indicators identified by the authors encompass the decrease of the energy consumption in water supply systems, as well as hydraulic head of pumps. Firstly, the indicators have been to be setup assembled to compare the regular operation conditions with its optimized conditions and, secondarily, the comparison of the energy is performance with the other systems. One of the most prominent indicators applied in such systems corresponds to the specific energy consumption (SEC), as shown by Equation (3). These parameters determined the energy the energy consumption to in which $C_{EL}$ (kW) designates the electricity consumption in a system and $Q$ $(\frac{m^3}{h})$ the water flow rate of a system.

$$\text{SEC} = \frac{C_{EL}}{Q_w} \tag{3}$$

The indicator has been modelled for the evaluation of energy consumption by a pumping system, in which the power of the pump depends on head demand of the water system. It also allows the comparison of the energy consumption between different scenarios as well as to compare different systems. For the comparison of the energy performance of different systems, Vilanova and Balestieri [44] also specifies an indicator in order to calculate the specific energy consumption for the same pump for a standard head of 100 m. This indicator is referred as the normalized specific energy consumption (NSEC).

Vilanova and Balestieri also proposes a KPI to evaluate the application of VSD's in pump motors. The Key Performance Indicator (KPI) corresponds to power demand part ($I_{OOBP}$) of the optimized pumping operation indicator, as defined by Equation (4). $P_{ELTOT}$ designates the total electrical power required by the pumping station and $P_{ELTOAT}$ the total electrical power current demand. The authors also mentioned a similar indicator, related to the supplied hydraulic head indicator, as defined by Equation (4). In Equation (5) $H_{AT}$ designates the target-value for average hydraulic head and $H_{OTMT}$ the optimized value for average hydraulic head.

$$I_{OOBP} = \frac{\min(P_{ELTOT})}{P_{ELTOAT}} \tag{4}$$

$$I_{CHD} = \frac{H_{AT}}{H_{OTMT}} \tag{5}$$

The energy performance of cooling systems is assessed through the determination of the cooling towers effectiveness. For open circuit cooling towers, this thermal efficiency is achieved according to Equation (6) [45], in which $T_{w,in}$ and $T_{w,out}$ correspond to the inlet and outlet water temperatures, respectively, and $T_{air,wb}$ the air wet-bulb temperature. For closed circuit cooling towers, its performance follows Equation (7), in which $T_{air,db}$ designates the air dry-bulb temperature. Considering the efficiency of open cooling tower around 70–75% [46], within this range, cooling towers may be considered efficient

in terms of heat transfer. Despite the inexistence of a precise reference to evaluate the performance of closed-circuit cooling towers, it is prudent to assume that this effectiveness is considerably higher.

$$\eta_{OCT} = \frac{T_{w,in} - T_{w,out}}{T_{w,in} - T_{air,wb}} \tag{6}$$

$$\eta_{CCT} = \frac{T_{w,in} - T_{w,out}}{T_{w,in} - T_{air,db}} \tag{7}$$

The potential savings for reduction in energy consumption for each sector and country were determined by applying the achieved shares of energy savings as a factor on the electricity consumption of each sector and country, according to Equation (8), in which $\varepsilon$ designates the achieved share of energy savings.

$$ELC_{potential} = \varepsilon \times ELC_{sector} \tag{8}$$

The potential savings for reduction in water consumption for each sector and country were determined by applying the achieved share of water savings as a factor on the Water consumption of each sector and country, according to Equation (9), in which $\omega$ designates the achieved share of water savings.

$$WC_{potential} = \omega \times WC_{sector} \tag{9}$$

## 4. Description of Case-Studies

The case-studies presented in this work are characterized by the circuit typology, correspondent industrial sector and country. Table 5 summarizes the details of each case-study (designated CS). The eleven case studies have been assembled in OpenModelica applying the WaterWatt library developed by the authors and presented in Iten et al. [47]. Following the previously introduced procedure, the current energy and water consumption have been estimated for all case studies and presented in Table 5.

**Table 5.** Description of WaterWatt case-studies (CS).

| Circuit | Designation | Industrial Sector | Origin | Water Flow Rate (m³/h) | Energy Consumption (MWh/year) |
|---|---|---|---|---|---|
| Rolling Mill cooling | CS1 | | Germany | 2200 | 4179 |
| Inductive furnace cooling | CS2 | | Germany | 65 | 183 |
| Blast furnace cooling | CS3 | Metal Industry | United Kingdom | 5543 | 19605 |
| Furnace gas washing | CS4 | | Germany | 900 | 10287 |
| Rebar rods and wire coils Quenching | CS5 | | Norway | 720 | 1743 |
| Manganese Blast Furnace | CS6 | | Norway | 684 | 1292 |
| Blast Furnace Gas Washing | CS7 | | | 255 | 1104 |
| Pharmaceutical Production Cooling | CS8 | Chemical Industry | Germany | 1322 | 1176 |
| Pulp transportation | CS9 | Paper and Pulp Industry | Portugal | 966 | 630 |
| Water treatment | CS10 | Food Industry | Portugal | 76 | 157 |
| Barometric condenser Cooling | CS11 | | | 1636 | 1375 |

Most of the case-studies circuits of this work are related to cooling circuit. The cooling circuits itself are divided into the open cooling typology, in which the heat transfer from water to the environment is processed in direct contact with another fluid, for instance, in cooling towers, and closed cooling typology, in which the referred heat transfer occurs in heat exchangers. Overall there are three case-studies for each one of these typologies. For instance, CS1, CS8, and CS11 correspond to open cooling circuits of rolling mills (steel industry), production of pharmaceuticals and barometric condensers (sugar industry), respectively. On the other hand, CS2, CS3, and CS6 circuits correspond to

closed cooling circuits of, inductive furnace (steel industry), blast furnace (steel production), and a furnace (manganese production), respectively. There are also two case-studies related to the gas washing circuit. The CS4 and CS7 correspond to gas washing circuits of basic oxygen furnaces. In addition, there is also the quenching circuit typology represented by CS5. The paper pulp transportation is represented in CS9 and the water treatment corresponds to CS10. Some of the abovementioned circuits are part of a same plant. For instance, CS1 and CS2 are part of the same stainless steel wire processing plant, CS6 and CS7 are part of the same manganese production plant and CS10 and CS11 are part of the same sugar production plant.

## 5. Results and Discussion

In this section, firstly improvement measures will be identified for each case study (Section 5.1). This is followed by its introduction into the modeling, according to an optimization methodology presented by Iten et al. [47]. The optimization methodology considers the operational requirements of the industrial plants by solving of optimization problems. The energy and water consumption reduction are considered as objective-functions and certain constraints are setup based on target-values related to operational requirements (operating temperature and pressure parameters of the circuits). A techno-economic assessment is presented in Section 5.2, focusing on the technical and economic aspects of the identified improvement measures for each case study. Such assessment allows to quantify energy and water savings and evaluate the viability of each measure, also allowing to prioritise these according to their payback time. To support the results achieved in Section 5.2, several KPI's are determined for the non-improved and improved scenarios of each case study. Finally, an analysis to the potential for water and energy efficiency improvement in each sector and country is performed in Section 5.4, taking into account the share of savings achieved in Section 5.2.

### 5.1. Description of Improvement Measures

The improvement measures have been identified with the aim of promoting energy and water savings in the current case studies. These measures are directly related with the operation of the equipment such as electric motors associated to pumps and cooling towers. Such equipment is common to most of the circuits and these technological measures reveal as great opportunities due to the adequacy of its application and the potential achievement in the reduction of consumption levels. Table 6 presents the improvement measures identified and analyzed in this work its description, the theoretical share of energy savings and the increases in its efficiency.

**Table 6.** Description of selected improvement measures. EEM: Energy Efficiency Improvement Measure.

| Measure | Name | Description | Share of Energy Savings and Increases in Efficiency |
|---|---|---|---|
| Installation of Variable Speed Drives (VSD's) in pump motors | EEM1 | It allows the automatic flow adjustment to the process needs, dynamically adjusting the pump rotation frequency to the optimal efficiency point. The replacement of the on/off cycle to continuous operation allows significant energy savings [48]. | 20–25% typical energy savings [49] |
| Installation of VSD's in cooling tower fans | EEM2 | It allows a dynamic adjustment of the airflow. The energy consumption associated to the operation of the fan is directly linked to the fan speed, therefore high energy savings can be achieved with adjustments decreasing the fan speed [48]. | |
| Replacement of IE1 Standard Efficiency motors to IE3 Premium Efficiency motors | EEM3 | The change of electric motors in pumping systems allow considerable energy savings, although a higher efficiency motor is more expensive than a conventional motor. Its lifespan is much longer, though, since heat losses are lower [49]. | 3–4% typical increase in efficiency [50] 8% maximum increase in efficiency [50] |
| Refurbishment of pumps | EEM4 | It consists in the mechanical cleaning and overhaul of a pump to approximately restore its initial functioning. In practice, such may be necessary due to loss of efficiency due to the degradation of the impeller and casing wear rings [51]. | 5–15% typical increase in efficiency [41] |

In addition to these measures, Iten et al. [47] also investigates the replacement of sand filters, commonly found in the water treatment units of the IWC. The principle behind the application of this measure is the decrease of the pressure loss along a circuit, which decreases the head demand of the pumps and consequently the power demand. The present work focuses on the measures translated directly into of the electricity consumption, therefore such measure as others related to the decrease of pressure losses in circuits are not considered in the present analysis. For each case study, the maximum number measures have been applied, in other words, the analysis included all the measures that are possible to be implement without compromising the circuit's operational requirements. Thus, each IWC will be analyzed considering on the application all these aggregated measures. For the EEM1 implementation, the energy savings have been determined considering the optimized rotational speeds of the pump motors which are in practice adjusted by the application of VSD's. The EEM2 implementation follows the same rationale but considering an optimized fan speed of the cooling towers. The energy savings have been determined considering the current scenario (before improvement) in which a circuit's pump and fans are operating at the nominal rotational speeds and the optimized scenario. The implementation of EEM3 and EEM4, respectively, consider the increase of the mechanical efficiency and hydraulic efficiency, respectively.

### 5.2. Techno-Economic Assessment

The techno-economic assessment of the implemented improvement measures has been performed to the energy and water savings accomplished in each circuit, considering the electric tariffs of each country case study. Such assessment aims to present the share of water and energy savings and the payback period for the selected measures. The energy consumption of each circuit was estimated on an annual basis, considering the operational time in hours per year of each circuit. Moreover, it was considered that the power consumption of each unit of the circuits presented a constant profile during its annual operational time. The payback time for each case study was determined according to Equation (1). Table 7 summarises the achieved energy and economic savings and payback period associated to the application of the aggregated measures in each case-study.

**Table 7.** Energy and economic savings and payback time for the application of aggregated measures.

| Sector | Circuit | Applied Measures | Energy Savings (MWh/year) | Share of Energy Savings (MWh/year) | Savings (€/year) | Payback Time (years) |
|---|---|---|---|---|---|---|
| Metal Industry | CS1 | EEM1 EEM2 EEM3 | 1061 | 25.3% | 208,791 | 1.1 |
| | CS2 | EEM2 EEM3 | 8 | 4.4% | 1570 | 5.7 |
| | CS3 | EEM2 EEM3 | 812 | 4.1% | 130,156 | 1.2 |
| | CS4 | EEM1 EEM2 EEM3 | 2416 | 23.5% | 475,214 | 0.5 |
| | CS5 | EEM1 EEM3 | 200 | 11.5% | 19,430 | 7.1 |
| | CS6 | EEM3 | 27 | 2.1% | 2624 | 12.1 |
| | CS7 | EEM2 EEM3 | 51 | 4.6% | 4960 | 5.2 |
| Chemical Industry | CS8 | EEM1 EEM2 EEM3 | 212 | 18.1% | 41,785 | 2.7 |
| Paper and Pulp Industry | CS9 | EEM1 EEM3 | 108 | 17.1% | 14,907 | 3.6 |
| Food Industry | CS10 | EEM1 EEM3 | 16 | 9.9% | 2149 | 5.7 |
| | CS11 | EEM2 EEM3 | 96 | 7.0% | 13,280 | 8.2 |

The investment costs for EEM1 and EEM2 were gathered from catalogues from WEG S.A. [52] and ThermoAir [53]. For EEM3, it was considered the pump motors catalogues of Siemens Industry [54] and Koupas [55]. The cost for EEM4 was estimated to be 2 full days of working hours of a maintenance technician with a monthly salary of 800 Euros.

The European industry considers an acceptable payback time between to as maximum of 3 years [56]. Analyzing Table 7, it is observable that the improvements for CS1, CS3, CS4, and CS8 are below the maximum acceptable value for payback time. For CS5, CS6, and CS11, the estimated payback time is sufficiently high so to claim that the application of all the measures simultaneously is not economically viable. Nonetheless, the estimated payback time depends not only on the achieved energy savings but also in the investment costs of the aggregated measures. Thus, despite all the measures, their implementation is possible simultaneously without surpassing the operational requirements of the plant, implementing the whole available measures may not be economically viable. Nonetheless, certain measures which in practice involve the replacement of equipment, such as the EEM3 and the substitution of the installed filters to ones associated to a lower pressure loss, may be justified, for instance, in the end of the equipment lifetime, independently of the associated payback time being attractive or not. Depending on the energy consumption levels, these figures have been presented for the different categories of energy consumptions case studies: low, average, and high. Figures 5–7 represent the actual energy consumption and the energy consumption after improved scenarios for each case-study.

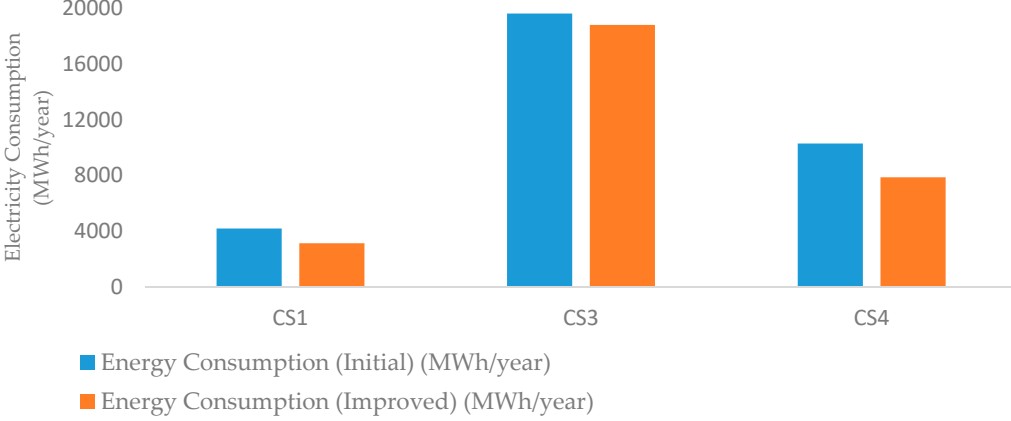

**Figure 5.** Electricity Consumption for the initial and improved operation of the industrial water circuits (IWC) (High consumption cases).

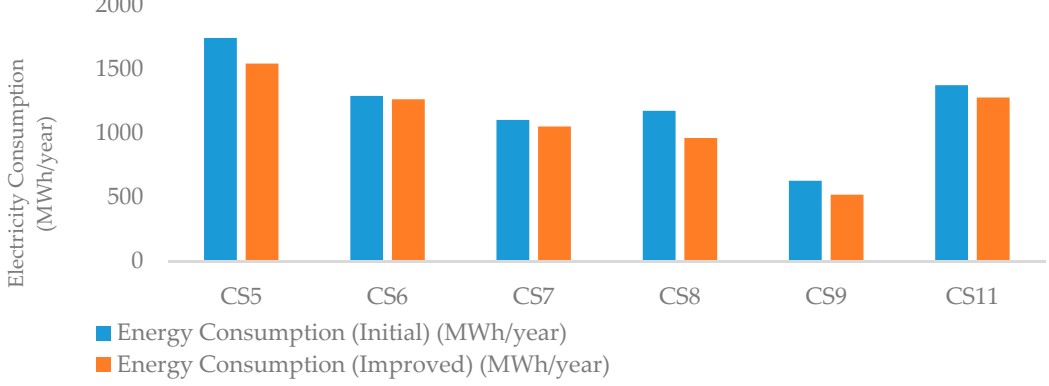

**Figure 6.** Electricity consumption for the initial and improved operation of the IWC (Average consumption cases).

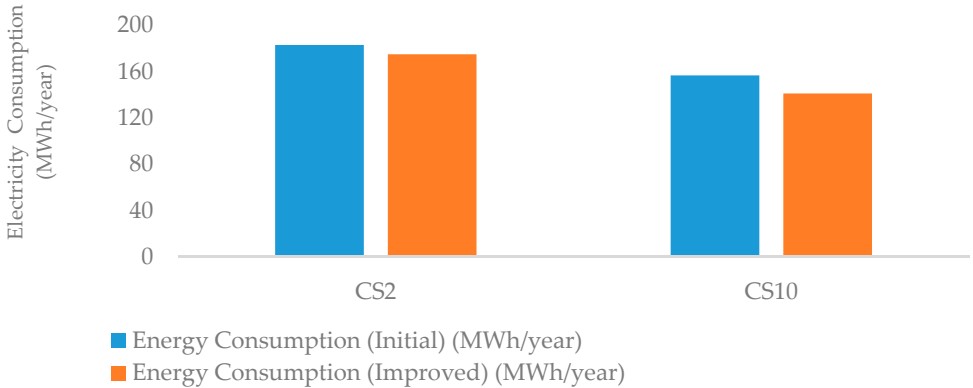

**Figure 7.** Electricity consumption for the initial and improved operation of the IWC (Low consumption cases).

In the case-studies with high electricity consumption (CS1 and CS4), appreciable savings have been achieved. On the other hand, the implementation of all the measures in CS3 has not allowed significant reductions of the energy consumption. The CS2 encompasses the circuit with the most significant energy consumption, so the improvement of energy efficiency would be, in principle, a greater concern. However, significant savings are not achieved because it corresponds to a constant water demand circuit, in which it is not possible to decrease the power of the pumps by adjusting the circuit to lower water demands.

For the case-studies with average energy consumption (CS5, CS6, CS7, CS8, CS9, and CS11), it is possible to claim a general fulfilment of the potential for energy efficiency improvement within circuits with variable water demand relative to the constant demand circuits. In particular, the energy performance of CS6 is a concern, since it is associated to a high consumption though the only improvement opportunity is related to the increase of the overall efficiency of pumps, not allowing significant energy savings with the implementation of VSD's. For the case-studies considered as low energy consumption it is possible to claim that the energy efficiency improvement of CS10 was successfully achieved. The same has however not been for CS2, in which EEM1, a measure associated to high opportunities for energy savings, was not implemented.

Overall, through the implementation of EEM1 it is possible to automatically adjust the circuit flow to lower water demands. Therefore, it is possible to achieve significant savings in water consumption by the application of this improvement measure. The decrease of water flow rate potentially allows the reductions of actual water consumption of the circuit, as exposed in Section 3. Table 8 summarizes the achievement of water savings achieved by the application of VSD's in pump motors.

**Table 8.** Water Savings achieved by the application of VSD's in pumps.

| Circuit | Water Savings (m$^3$/h) | Share of Reduction in Water Consumption (%) |
|---------|-------------------------|---------------------------------------------|
| CS1 | 106 | 4.81 |
| CS4 | 45 | 5.00 |
| CS5 | 22 | 3.06 |
| CS8 | 97 | 7.34 |
| CS9 | 3 | 0.58 |

*5.3. Determination of Key Perfomance Indicator (KPI)*

Water and energy efficiency may also be evaluated through the determination of specific indicators designated as key performance indicators (KPIs). Such KPIs allow to compare different circuits and evaluate its efficiency in terms of the water–energy nexus.

5.3.1. KPI for Pumping System and Overall Circuit Operation

The assessment of the energy performance for the circuit operation is performed according to Equation (3). For each scenario and each circuit, this indicator allows the assessment of the energy efficiency improvement of a circuit by understanding how much energy is consumed to transport one cubic meter of water. Table 9 shows that SEC decreases between the non-improved circuits to the improved scenarios for the same demand of water. These values reflect the results for the energy savings previously presented.

**Table 9.** Specific energy consumption for the non-improved and improved scenarios for each circuit.

| Circuit | SEC (kWh/m$^3$) | SEC (Improved) (kWh/m$^3$) |
|---------|-----------------|----------------------------|
| CS2 | 0.426 | 0.408 |
| CS3 | 0.268 | 0.257 |
| CS6 | 0.214 | 0.210 |
| CS7 | 0.496 | 0.473 |
| CS11 | 0.286 | 0.251 |

The assessment of the energy performance of the pumping systems, it is calculated the normalized specific energy consumption (NSEC) for a reference pump head of 100 m, as proposed by Vilanova and Balestieri [44]. In the scope of this study, the NSEC is defined and calculated for each individual pump group of each circuit. The aim here is the comparison of the energy performance between different circuits by comparing the specific energy consumption of all the pump groups, rather than comparing the energy performance of different pump groups within the same circuit. Table 10 presents the NSEC values for each pump group of each circuit.

**Table 10.** Normalized specific energy consumption (NSEC) for each one of the pump groups of each circuit.

| | NSEC (kWh/m$^3$) | | |
|---|---|---|---|
| | **Pump Group 1** | **Pump Group 2** | **Pump Group 3** |
| CS1 | 0.362 | 0.426 | 0.300 |
| CS2 | 0.616 | 0.595 | |
| CS3 | 0.336 | 0.381 | |
| CS4 | 0.661 | 0.985 | |
| CS5 | 0.355 | | |
| CS6 | 0.350 | | |
| CS7 | 0.480 | | |
| CS8 | 0.333 | | |
| CS9 | 0.447 | 0.512 | 0.544 |
| CS10 | 0.486 | 0.712 | |
| CS11 | 0.564 | 0.550 | |

Observing Table 10, it is possible to establish a distinction between circuits with a pumping system with high and low energy performance achieved. The NSEC achieved determine that CS1, CS3, CS5, CS6, and CS8 have a high energy pumping performance (less than 0.450 kWh/m$^3$), while CS2 and CS4 correspond to low performance group (above 0.600 kWh/m$^3$). Despite being a circuit with high energy consumption, CS3 is associated to low values of NSEC, thus to high energy performance of the pumps. Such observation is due to the high-water flow rate of the circuit, which is responsible for the high energy demands, and the reason for high energy consumption rather than low overall efficiencies of the pumps or high pump heads. In order to assess the implementation of the proposed improvement measures, the indicators $I_{OOBP}$ and $I_{CHD}$ have been determined according to Equations (4) and (5). Table 11 summarizes the obtained results for each circuit.

**Table 11.** Optimized pumping operation and supplied hydraulic head indicator for variable water demand circuits.

| Circuit | $I_{OOBP}$ (%) | $I_{CHD}$ (%) |
|---------|----------------|----------------|
| CS1  | 98.1 | 98.6 |
| CS4  | 96.0 | 97.1 |
| CS5  | 83.0 | 89.1 |
| CS8  | 93.7 | 95.8 |
| CS9  | 87.4 | 91.9 |
| CS10 | 79.3 | 86.9 |

The $I_{OOBP}$ and $I_{CHD}$ indicators for all the circuits were calculated for the same target, which is a 25% energy savings owing to the installation of VSD's in pumps. The target-values of energy consumption are, thus, calculated as 75% of the initial energy consumption. The comparison of the achieved results allows to identify the circuits in which the plant operational requirements are more restrictive in relation to the achieved energy savings. The $I_{OOBP}$ for CS1, CS4, and CS8 are relatively high indicating that the operational requirements in these circuits are not restrictive for the implementation of the improvement measure.

### 5.3.2. KPI for the Cooling System

The assessment of the energy performance of the cooling systems is performed through the calculation of the cooling towers effectiveness, according to Equations (6) and (7). Although the focus of this work is the study on the implementation of improvement measures for directly evident on the reduction of electricity consumption. Such effectiveness is indirectly considered (Table 12). For instance, the reduction of the electric consumption of the cooling tower fan by the reduction of the rotational speed, will decrease the air flowrate and consequently there will be an impact on the heat transfer between the air and the water. Overall, such analysis may evaluate if the current operation of the cooling towers in the circuit is adequate to cooling capacity required for the operation of the circuit, considering the temperature requirements. Hence, the operation of cooling towers with low thermal efficiency may turn adequate by conjoining a measure such as Energy Efficiency Improvement Measure 2 (EEM2).

**Table 12.** Cooling tower thermal efficiency for cooling and gas washing circuits.

| Circuit | Cooling Tower Arrangement | Thermal Efficiency (%) |
|---------|---------------------------|------------------------|
| CS1  | Open Circuit   | 17.7 |
| CS2  | Closed Circuit | 22.6 |
| CS3  | Open Circuit   | 53.2 |
| CS4  | Open Circuit   | 73.8 |
| CS7  | Open Circuit   | 63.4 |
| CS8  | Open Circuit   | 27.0 |
| CS11 | Open Circuit   | 66.5 |

Within the circuit containing open cooling towers, as it may be observed from Table 11, CS4 is the only case in which this effectiveness is within the range considered that the heat transfer between air and water is efficient.

### 5.4. Potential for Water and Energy Efficiency Improvement Measures at European Level

As claimed by Saidur [41] and Almeida et al. [43], the electric motors represent specific shares of energy consumption within each of the industrial sectors. The exact values for these shares are summarized in Figure 4 and Table 4. Based on the results shown in Table 7, the circuit improvement allows high energy savings, that corresponds to a maximum of 25.3% of reduction in electricity

consumption in a circuit. In this ground, despite being considered secondary to the production processes, IWC represent a large potential of the overall the energy savings at industries. Hence, the savings achieved in typical industrial water circuits presented in this study may be projected to each industrial sector of different countries. The same analysis is presented to the water consumption. Such survey analysis of the water and energy consumptions enables to frame the achieved results for circuits optimization within the water–energy nexus.

### 5.4.1. Potential for Energy Efficiency Improvement at EU Level

The energy savings achieved for each representative circuit are projected in each sector and in each country. For each country, the case studies which major shares of consumption reductions are considered, in order to represent the maximum energy savings in each sector. The levels of average electricity consumption in German manufacturing industry are more significant compared to other Member States, such as Portugal, as well as to the average levels of the European Union, as observed Figure 2. The potentials for reduction in energy consumption for each sector and country were determined by applying a share of energy savings, attending to Equation (8).

The techno-economic assessment of this work includes a set of case studies in Germany corresponding to the Iron and Steel Industry sector, namely CS1, CS2 (at the same industry), and CS4. The plant of CS1 and CS2 has shown the most significant share of energy savings for the implemented measures. The chemical industry sector in Germany is represented by CS8, which also shows a high share of energy savings. Considering that the average consumption levels presented on Section 2.2 correspond to the total electricity consumption in each sector and country, for the determination of potential savings in the current status of the different countries is necessary to estimate the average energy consumption by the electric motors in these specific sectors. The energy consumption for the metal industry sector in Germany has been determined and presented in Figure 2. The share of electric motor energy consumption corresponds to 66.3% [43] as observed in Table 4. For the chemical industry, a share of 71.9% [43] has been considered. The Figure 8 represents the projected potential savings of energy savings for each industrial sector in German industry.

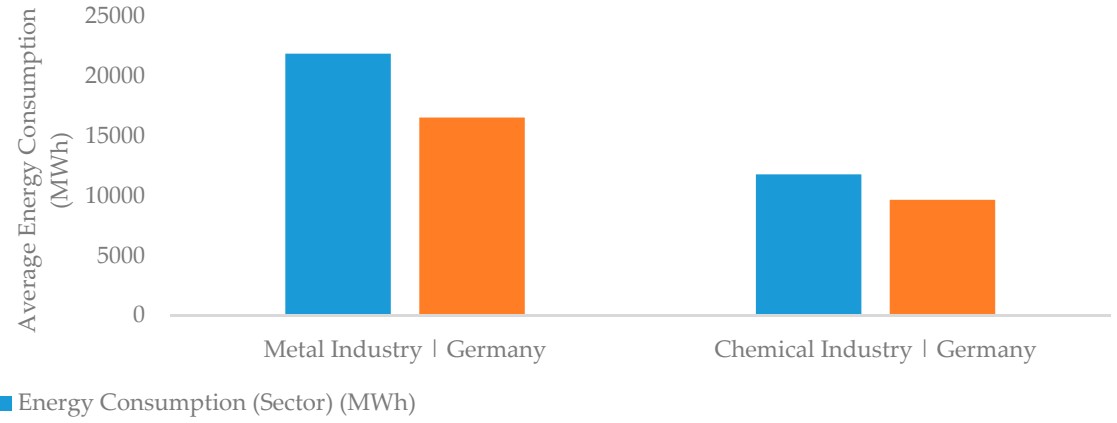

**Figure 8.** Potential savings in energy consumption for each approached industrial sector in Germany (data adapted from [35,36,43]).

The same analysis was perfomed for the paper and pulp Industry, represented by CS9 and the food industry, represented by the CS10 and CS11, respetively. For the first case, a share of energy consumption of 75.1% [43] has been considered relatively to the consumption levels presented in Figure 2. For the latter, a share of 89.8% [43] has been considered. The Figure 9 represents the projected potential savings of energy for the Portuguese industries.

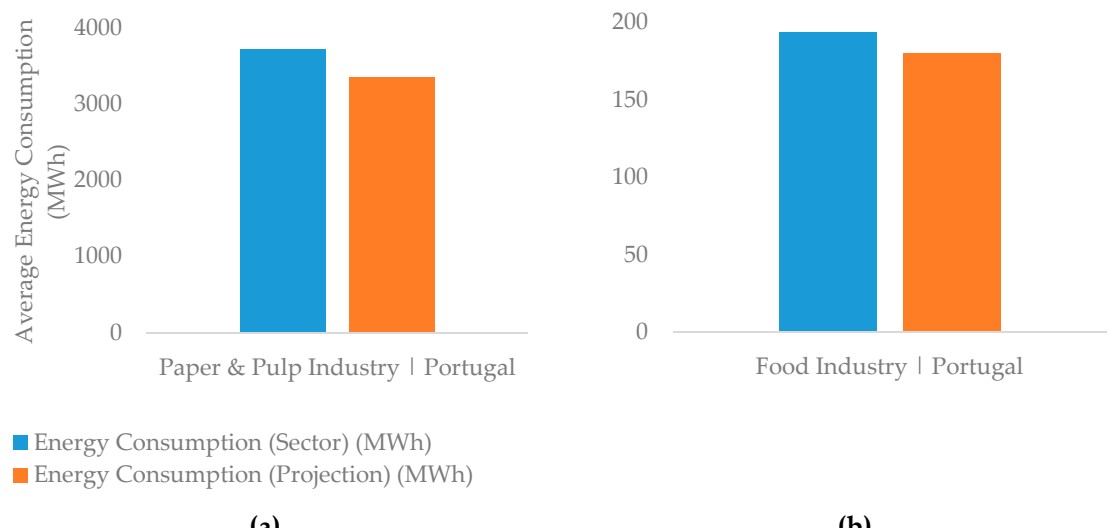

**Figure 9.** Potentials savings in energy consumption for each approached industrial sector in Portugal. (**a**) paper & pulp industry, (**b**) food industry.

The metal industries in the UK and Norway are represented by CS3 and CS5, respectively. The potential savings in energy consumption are determined considering the average European Union levels, due to, as previously mentioned, the inexistence of data namely of the number of enterprises in both countries and therefore disabling the estimation of the average energy consumption in these two countries. The share for the UK was considered as 66.3%, the same as for the the Iron and Steel Industry in Germany. For Norway, a share of 59.9% has been considered, corresponding to the Other Industries, as presented in Table 4 [43]. This assumption has been made as this particular plant is not encompassed by the Iron and Steel sector. The Figure 10 represents the projected savings of energy for the UK and Norwegian industrial sectors.

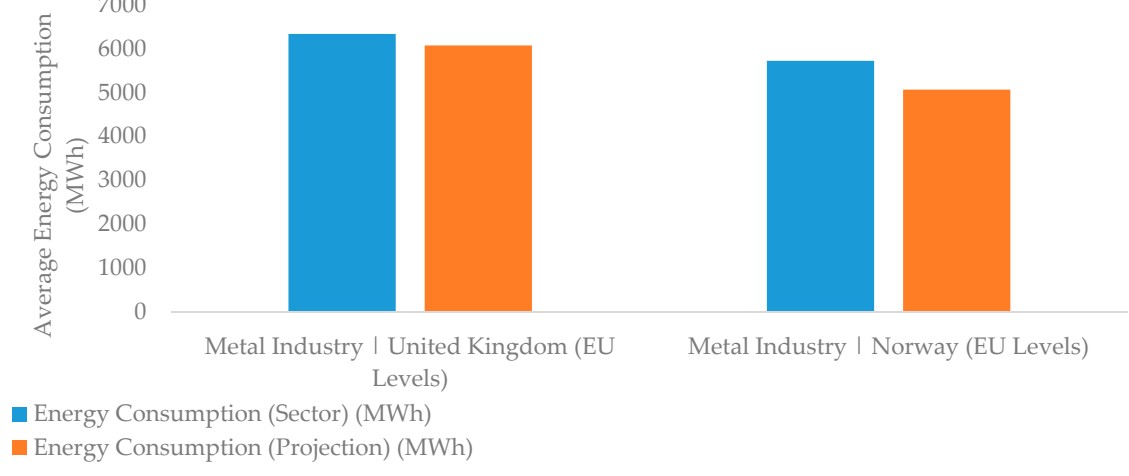

**Figure 10.** Potentials for savings in energy consumption for each approached industrial sector in the United Kingdom and Norway (based on the average levels of the European Union).

The results for the energy savings for both circuit level and the average sectorial level (calculated per number of enterprise) are presented on Table 13.

**Table 13.** Potential energy savings in each industrial sector at EU level.

| Sector\|Member State | Energy Savings (Circuit Level) (MWh/year) | Energy Savings (Sectorial Level) (TWh) |
|---|---|---|
| Metal Industry\|Germany | 1069 | 5340 |
| Metal Industry\|United Kingdom (EU Levels) | 812 | 262 |
| Metal Industry\|Norway (EU Levels) | 200 | 655 |
| Chemical Industry\|Germany | 212 | 2130 |
| Paper and Pulp Industry\|Portugal | 108 | 369 |
| Food Industry\|Portugal | 112 | 14 |

Considering the number of enterprises in each sector and country, it is possible to setup potential savings relative to the total electricity consumption in a specific sector and country. In Germany, for the metal industry, a reduction of energy consumption corresponds to 1069 MWh/year. For the chemical industry, it corresponds to 212 MWh/year. For both industries, the projected potential corresponds to a total of 7.0 TWh of energy savings. Considering the energy efficiency targets delineated by the NEEAP for Germany of about 307 TWh [57] between 2008 and 2014, it is observed that improvement measures implementation in both metal industry and chemical industry in Germany can account for approximately 2.3% of that target. This perspective indeed reflects the actual progress in terms of energy efficiency in Germany.

In Portugal, a 108 MWh/year of energy savings represent a 211 GWh total energy savings for the paper and pulp sector. For the food sector, 112 MWh/year of energy savings would represent a 126 GWh reduction for the total of the sector. Considering such energy savings and the target delineated in the NEEAP, namely a reduction of 1.6 TWh [58] between 2008 and 2016 it is observed that the achieved savings, would represent a 6.3% and 12.5% of that target, respectively for the paper and pulp and food industries. This prospect also reveals the actual appreciable progress in energy efficiency in this country.

In the cases of Norway and UK, as the projected potentials have been estimated at EU level, the comparison of the achieved savings is performed with the European Union targets which corresponds to 1087 TWh [59]. For Norway in specific, the energy savings correspond to a total reduction of 12.6 TWh in the metal industry, which represents 1.2% of the target for EU. For the UK, the energy savings correspond to 5 TWh in the same sector, representing 0.5% of the target of EU.

Overall, the potential savings achieved for the metal industry in Germany are much higher than in Norway and the UK. In the UK, the potential for energy efficiency improvement is rather less optimistic. Such may reflect that the implementation of energy efficiency is in the need of a promotion, as actually evidenced [23].

5.4.2. Potential for Water Efficiency Improvement at Sector Level

The potential water consumption reduction for each sector and country were determined by applying a share of water savings, attending to Equation (9). The techno-economic assessment presented in Section 5.2 has shown the improvement measures do not only allow considerable energy savings but also savings in water consumption. This has been achieved considering the application of VSD's in pumps motors in order to adjust to lower water demands. Hence, the circuits water flow rate also decreases. As part of the framework of water–energy nexus, the shares of water savings have also been projected into potential savings for each sector and country. Such analysis has been performed considering the results presented in Table 7 and the consumption levels represented in Figure 3. The potential for water efficiency improvement has been performed in a similar manner as the energy efficiency improvement case. However, in this analysis the circuits with variable water demand have been considered (CS1, CS4, CS5, CS8, CS9, and CS10).

In this analysis, it is admitted the hourly basis used for the determination of water savings is proportionate to the annual basis used for each sector and country. The water savings for the average

sectorial level (calculated per number of enterprise) are presented on Table 14. Figures 11 and 12 represent the projected potentials for the water savings for each sector and country.

**Table 14.** Potential savings in water consumption in each industrial sector at EU level.

| Sector\|Member State | Savings in Water Consumption (Sectorial Level) (1000 m$^3$/year) |
|---|---|
| Metal Industry\|Germany (CS1 and CS2) | 21.4 |
| Metal Industry\|Norway (EU Levels) (CS5) | 10.8 |
| Chemical Industry\|Germany (CS8) | 28.9 |
| Paper and Pulp Industry\|Portugal (CS9) | 0.6 |

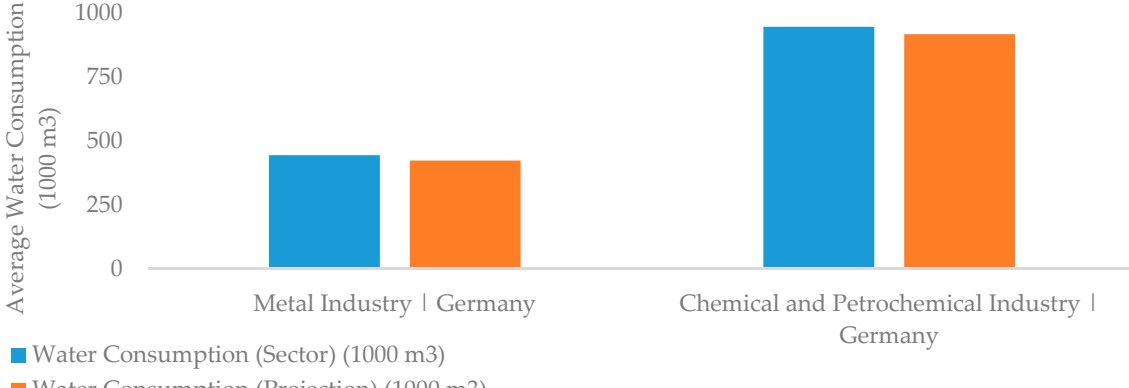

■ Water Consumption (Sector) (1000 m3)
■ Water Consumption (Projection) (1000 m3)

**Figure 11.** Potentials savings in water consumption for each approached industrial sector in Germany.

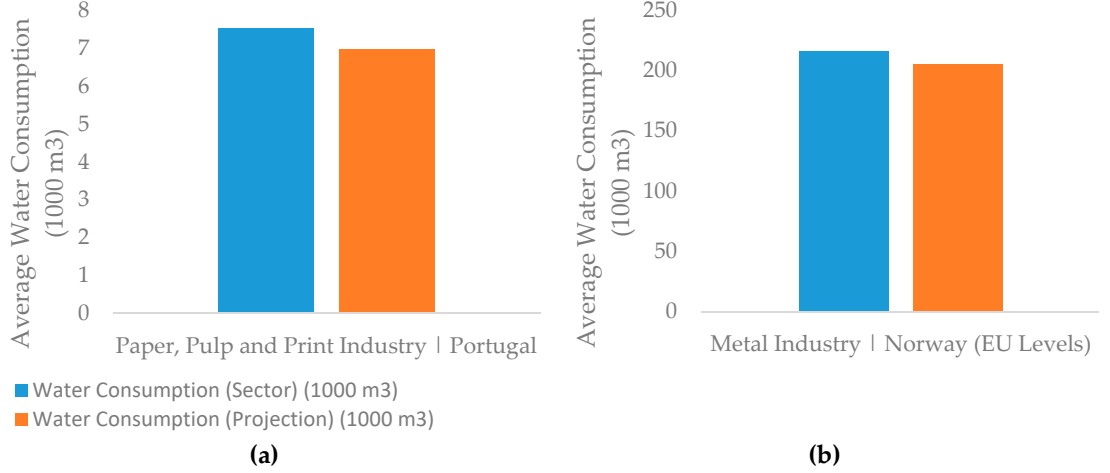

■ Water Consumption (Sector) (1000 m3)
■ Water Consumption (Projection) (1000 m3)

**(a)**  **(b)**

**Figure 12.** Potentials savings in water consumption for the paper & pulp industry in Portugal (**a**) and for the metal industry in Norway (**b**) (based on the average levels of the European Union).

From Table 14 is possible to verify that a reduction of the flow rate corresponds to 106 m$^3$/h in the Metal Industry circuit in Germany corresponds to a projected potential savings of $21.4 \times 10^3$ m$^3$ of water in the sector. For the same sector considering the EU level, in Norway such prospect it corresponds to 22 m$^3$/h reduction of the water flow rate corresponds to a $10.8 \times 10^3$ m$^3$ reduction in the sector. Considering that the level of water consumption in Norway is inferior to Germany, it is possible to claim that it is possible to achieve considerable water savings in Norway compared to the prospect for Germany. For the chemical industry in Germany, a water flow rate reduction corresponds to 97 m$^3$/h and respectively for $28.9 \times 10^3$ m$^3$ for this sector. Since the water consumption level for the chemical industry in Germany is much higher than for the metal industry, such prospect has a less positive impact. For the paper and pulp industry in Portugal, the water consumption reduction

corresponds to 3 m$^3$/h and the correspondent circuit accounts for $0.6 \times 10^3$ m$^3$/year in this sectorial level. Since Portugal is associated to low average water consumption comparatively to the other Member States, it would be expected that projected water savings would be lower, which is effectively observed by this result.

## 6. Conclusions

This work presents several improvement measures for water circuits in order to reduce the overall water and energy consumption in industries. Such savings have been projected considering the current water and energy consumption in the EU Member States and per specific sectors of the manufacturing industry. The achievements have been contextualized in the concept of interdependencies between water and energy, namely the water–energy nexus.

A techno-economic assessment has shown the potential of the improvement measures on energy savings, with a maximum share of reduction by 25.3% related to the circuit presenting less restrictive constraints. This has also been translated into the reduction in water flow rate corresponding to up to 106 m$^3$/h.

The potential savings projected for the current water and energy demand in different EU members and industrial sectors have also been presented. It evidenced that Germany, Portugal and Norway present a higher potential to reduce both water consumption and the total energy consumption, namely by the optimization of electric motors operation. Considering the target for energy consumption of each Member State, it was observed that the potential energy savings may account for considerable targets of the final energy consumption by: 2.3% for both the metal and chemical sectors in Germany; 6.3% and 12.5% for the paper and pulp and food industries in Portugal, respectively; and 1.2% for the metal industry in Norway. This reflects the overall appreciable progress of the energy efficiency improvement in these Member States. Contrariwise, in the UK, these prospects have been far less appreciable. In this country, the potential energy savings for the metal sector represents 0.5% of the final energy consumption target-value.

As envisioned by the water–energy nexus, it is observed that the water circuits associated to a high energy consumption are also the ones associated to high water flow rates. Such was especially verified through the elaborated potential savings analyses. In these analyses, energy efficiency results are in accordance to the water efficiency results. Therefore, the tendencies related to energy efficiency improvement in such circuits correspond to a similar trend as water efficiency improvement.

**Author Contributions:** M.C.O. conceived and designed the case studies; M.C.O. performed the modeling; M.I. and H.M. analyzed the data; M.C.O, M.I., H.A.M wrote the paper; M.I and H.A.M and J.M Revised the paper.

**Funding:** This project has received funding from the European Union's Horizon 2020 research and innovation programme under grant agreement "No 695820".

**Conflicts of Interest:** The authors declare no conflict of interest.

## Nomenclature

**Index**

| | |
|---|---|
| *air* | Air |
| *db* | Dry-bulb |
| *potential* | Projected potential |
| *in* | Inlet |
| *Installation* | Installation |
| *Inv* | Investment |
| *Maintenance* | Maintenance |
| *OCT* | Open Circuit Cooling Tower |
| *CCT* | Closed Circuit Cooling Tower |

| | |
|---|---|
| *res* | Residence |
| *sector* | Sector |
| *w* | Water |
| *wb* | Wet-bulb |

**Abbreviations**

| | |
|---|---|
| BAT | Best Available Technologies |
| CS | Case study |
| EEM | Energy Efficiency Improvement Measure |
| EU | European Union |
| KPI | Key Performance Indicator |
| min | Minimal |
| NEEAP | National Energy Efficiency Action Plan |
| UN | United Nations |
| VSD | Variable Speed Drives |

**Parameters**

| | |
|---|---|
| $C_{EL}$ | Power consumption of a water system (kW) |
| $C_{Inv}$ | Investment cost of a technological measure (€) |
| ELC | Electricity Consumption (MWh) |
| $H_{AT}$ | Average pump hydraulic head average value (m) |
| $H_{OTMT}$ | Average pump hydraulic head target-value (m) |
| $I_{CHD}$ | Supplied Hydraulic head indicator (%) |
| $I_{OOBP}$ | Optimized pumping operation indicator demand part (%) |
| NSEC | Normalized specific electricity consumption (kWh/ m$^3$) |
| $P_{ELTOT}$ | Total electric power required by a pumping system (kW) |
| $P_{ELTOAT}$ | Total electric power current demand (kW) |
| PB | Payback period (year) |
| $Q_w$ | Water volumetric flow rate (m$^3$/h) |
| SEC | Specific electricity consumption (kWh/ m$^3$) |
| Sav | Economic savings associated to an improvement measure (€/year) |
| T | Water temperature (°C) |
| $t_{res}$ | Residence time (h) |
| $V_w$ | Water volume (m$^3$) |
| WC | Water Consumption (10$^3$ m$^3$) |
| $\eta_{OCT}$ | Open circuit cooling tower thermal effectiveness (%) |
| $\eta_{CCT}$ | Closed circuit cooling tower thermal effectiveness (%) |
| $\varepsilon$ | Share of Energy Savings (%) |
| $\omega$ | Share of Water Savings (%) |

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
