# Peer review of "Water–Energy Nexus in Typical Industrial Water Circuits"

_water, doi:10.3390/w11040699_

Round 1

Reviewer 1 Report

It is an interesting and relevant study, but is quite poorly written.

1.       It would be better to separate “Materials and Methods” and “Results and Discussion” instead of lumping them together.  

2.       The work does not seem to have a good awareness of the water-energy nexus and industrial energy/water efficiency literature. While it is good to include a lot of non-academic literature references for the policy context, only a few academic papers have been referenced. It makes it hard to justify the contribution of this work. Are there works also quantify the potential energy and water saving from efficiency measures in the EU context?

3.       “Water circuits”/ “circuits” are not defined anywhere. As far as I am aware of, it is not a common term in the water-energy nexus context. Does it mean “process component” / “unit operation”? Some of the examples do not seem to have a water circulation nature.

4.       Numerous unclear sentences and typos. Suggest carefully going through it. Just list a few: Lines 50-51, “to improve the energy efficiency of water and energy”, Lines 544-545

5.       Line 13 not just limit to the electricity supply system

6.       Refs [1] and [2] are wrong?

7.       Figure 1 is too small.

8.       Figure 2 the unit of y-axis should be annual average electricity consumption per enterprise?

9.       Figure 3 “Water Energy Consumption”? What is the unit “dam3”?

10.   Check the display of all other figures.

11.   Consider combining some of the figures. There are too many figures now.

12.   “Electric energy”, why not just “electricity”?

13.   For the economic assessment, what is the basis of electricity cost? No consideration of discount rate?

14.   Forecast and projection are not the same. And they usually have a future time frame. Is the work more about potential savings from the measures?

Author Response

The provided comments are much appreciated and will improve significantly the manuscript.

Amendments have been done accordingly with the comments provided by the reviewer. All changes have been highlighted throughout the manuscript.

1.      It would be better to separate “Materials and Methods” and “Results and Discussion” instead of lumping them together.  

The authors have separated accordingly: Chapter 3 denominated as “Materials and Methods” and Chapter 5 as "Results and Discussion".

2.      The work does not seem to have a good awareness of the water-energy nexus and industrial energy/water efficiency literature. While it is good to include a lot of non-academic literature references for the policy context, only a few academic papers have been referenced. It makes it hard to justify the contribution of this work. Are there works also quantify the potential energy and water saving from efficiency measures in the EU context?

The authors have addressed references [12], [16], [17], [18], [19] and [20] in order to cover more academic papers as well works related to potential energy and water savings in EU context.

3.       “Water circuits”/ “circuits” are not defined anywhere. As far as I am aware of, it is not a common term in the water-energy nexus context. Does it mean “process component” / “unit operation”? Some of the examples do not seem to have a water circulation nature.

Definition has been added in the manuscript (Lines 79- 82).

4.      Numerous unclear sentences and typos. Suggest carefully going through it. Just list a few: Lines 50-51, “to improve the energy efficiency of water and energy”, Lines 544-545

This has been amended.

5.      Line 13 not just limit to the electricity supply system

This has been amended. 

6.      Refs [1] and [2] are wrong?

This has been amended.

7.      Figure 1 is too small.

This imaged has been increased for a better clarity.

8.      Figure 2 the unit of y-axis should be annual average electricity consumption per enterprise?

This legend has been accordingly amended in Figure 2.

9.      Figure 3 “Water Energy Consumption”? What is the unit “dam3”?

The legend y-axis has been changed and the unit has been amended to 1000 m3.

10.   Check the display of all other figures.

All figures have been checked and amended accordingly.

11.   Consider combining some of the figures. There are too many figures now.

Figs. 12 and 13 have been merged.

12.   “Electric energy”, why not just “electricity”?

This has been amended along the manuscript accordingly.

13.   For the economic assessment, what is the basis of electricity cost? No consideration of discount rate?

Details have been added and highlighted in the manuscript.

14.   Forecast and projection are not the same. And they usually have a future time frame. Is the work more about potential savings from the measures?

Indeed, the work is about potential savings. This has been amended accordingly and  adopted the highlighted in the manuscript.

Reviewer 2 Report

Minor observations

Lines 44-51: Why to address reference [7] before reference [6]? Re number and change in references also.

Think to use Greek letters normal (not italic).

Good work! Congratulations!

Author Response

Amendments have been done accordingly with the comments provided by the reviewer. All changes have been highlighted throughout the manuscript.

1.       Lines 44-51: Why to address reference [7] before reference [6]? Re number and change in references also.

This has been amended accordingly.

       2. Think to use Greek letters normal (not italic).

This has been amended.

Round 2

Reviewer 1 Report

The revision is okay. Just spotted some typos. Please proofread carefully.

Line 124 Thiede

Line 505 potential